# Lung Metabolomics Profiling of Congenital Diaphragmatic Hernia in Fetal Rats

**DOI:** 10.3390/metabo11030177

**Published:** 2021-03-18

**Authors:** Maria del Mar Romero-Lopez, Marc Oria, Miki Watanabe-Chailland, Maria Florencia Varela, Lindsey Romick-Rosendale, Jose L. Peiro

**Affiliations:** 1Center for Fetal and Placental Research, Division of Pediatric General and Thoracic Surgery, Cincinnati Children’s Hospital Medical Center (CCHMC), Cincinnati, OH 45229, USA; maria.romero.lopez@cchmc.org (M.d.M.R.-L.); marc.oria@cchmc.org (M.O.); maria.florencia.varela@cchmc.org (M.F.V.); 2Perinatal Institute, Division of Neonatology, Cincinnati Children’s Hospital Medical Center, Cincinnati, OH 45229, USA; 3Department of Surgery, College of Medicine, University of Cincinnati, Cincinnati, OH 45267, USA; 4NMR-based Metabolomics Core, Division of Pathology and Laboratory Medicine, Cincinnati Children’s Hospital Medical Center, Cincinnati, OH 45229, USA; Miki.Watanabe@cchmc.org (M.W.-C.); Lindsey.romick-rosendale@cchmc.org (L.R.-R.)

**Keywords:** CDH, metabolism, metabolomics, lung, fetus, development

## Abstract

Congenital diaphragmatic hernia (CDH) is characterized by the herniation of abdominal contents into the thoracic cavity during the fetal period. This competition for fetal thoracic space results in lung hypoplasia and vascular maldevelopment that can generate severe pulmonary hypertension (PH). The detailed mechanisms of CDH pathogenesis are yet to be understood. Acknowledgment of the lung metabolism during the in-utero CDH development can help to discern the CDH pathophysiology changes. Timed-pregnant dams received nitrofen or vehicle (olive oil) on E9.5 day of gestation. All fetal lungs exposed to nitrofen or vehicle control were harvested at day E21.5 by C-section and processed for metabolomics analysis using nuclear magnetic resonance (NMR) spectroscopy. The three groups analyzed were nitrofen-CDH (NCDH), nitrofen-control (NC), and vehicle control (VC). A total of 64 metabolites were quantified and subjected to statistical analysis. The multivariate analysis identified forty-four metabolites that were statistically different between the three groups. The highest Variable importance in projection (VIP) score (>2) metabolites were lactate, glutamate, and adenosine 5′-triphosphate (ATP). Fetal CDH lungs have changes related to oxidative stress, nucleotide synthesis, amino acid metabolism, glycerophospholipid metabolism, and glucose metabolism. This work provides new insights into the molecular mechanisms behind the CDH pathophysiology and can explore potential novel treatment targets for CDH patients.

## 1. Introduction

Congenital diaphragmatic hernia (CDH) is characterized by a defect in the diaphragm natural fusion early in gestation, leading to herniation of abdominal viscera into the thoracic cavity, causing competition for space in the fetal chest. This scenario results in severe pulmonary hypertension (PH) by altering vascular branching, histology, and lung hypoplasia [1]. The parenchymal/pulmonary vasculature and heart development are impeded due to an early embryogenic alteration. Moreover, the pathophysiology of the disease is still not well understood.

The CDH incidence is 1.93/10,000 births in North America, with an overall 45.89% mortality in the first year of life [2]. Half of the cases, approximately, are associated with other congenital malformations [3,4,5]. The survivors face considerable long-term morbidities, such as respiratory problems, nutritional issues, neurodevelopmental delays, hernia recurrence, and orthopedic complications, needing a multidisciplinary approach [2,6]. Pathological gastro-esophageal reflux disease and the presence of hiatal hernia in patients who have undergone CDH repair are both known long-term complications [7,8].

PH is a significant cause of mortality in CDH patients. PH is driven by hypoplastic and abnormal pulmonary vasculature [1,9]. The main pathways implicated in this problem are the retinoic acid, nitric oxide, vascular endothelial growth factor, and endothelin pathways [10,11,12,13,14,15]. Fetal left heart structures are usually smaller in CDH [16,17,18] and correlate with fetal lung hypoplasia [19], probably related to compression during fetal development. The cardiac malposition due to the heart’s displacement can cause a redistribution of the flow from the inferior vena cava that goes mainly to the pulmonary artery instead of the left ventricle [20]. Hypoplastic CDH lungs have a decrease in the pulmonary flow, being worse on the ipsilateral side [21,22,23,24]. Decreased pulmonary blood flow is sufficient to cause lung parenchymal hypoplasia [25]. Also, the fetal lungs are compressed by the abdominal content, causing increased pulmonary vascular resistance and decreasing lung flow.

The treatment of CDH- associated PH remains one of the most challenging therapies. Although multiple histological and molecular abnormalities are causing pulmonary hypertension as previously described, most of the current treatments target only one of the potentially involved mechanisms. Recent investigations have identified metabolic changes in PH with a shift from oxidative phosphorylation to glycolysis (Warburg effect) [26]. The metabolic dysregulation in PH goes beyond the Warburg effect, including alterations in the glutaminolysis, fatty acid handling, and pentose phosphate pathway [27].

The study of lung metabolism has made significant progress in the last decades [28], but still, there is a lack of knowledge specifically in CDH lung metabolism. Nuclear magnetic resonance (NMR)-based metabolomics is a reliable and untargeted method for detecting metabolic signature changes under stress, providing a comprehensive description of complex diseases’ biochemical changes.

We investigated fetal lung metabolism of the well-established nitrofen CDH model in rats using Nuclear magnetic resonance (NMR) spectroscopy in the current project. The knowledge of specific metabolic profiles of CDH can help understand this disease’s pathophysiology better, identify potential biomarkers, and develop novel and more efficient therapies.

## 2. Results

### 2.1. CDH Fetal Lungs Have a Different Metabolic Profile

To determine the metabolites involved in the pathological process of CDH resulting from nitrofen maternal administration at E9.5 of gestation in pregnant rats, total lung of fetuses in each experimental group: pups exposed to nitrofen who developed CDH (NCDH), pups exposed to nitrofen who did not develop CDH (NC), and control pups from dams that received olive oil vehicle (VC), were analyzed using an NMR-based metabolomics approach. First, overall, 64 metabolites’ abundance was assessed by principal component analysis (PCA) to observe the data’s overall unsupervised pattern. The PCA scores plot showed clear separations between the three groups onto the first two principal components (PCs), accounting for 69.7% of the total variance (51.8% and 17.9%, respectively) of the PCA scores plot (Appendix A). Based on the loadings plot, the metabolites responsible for this separation in the PC1 direction were lactate, adenosine monophosphate (AMP), uridine 5′-diphosphate (UDP), alanine, hypotaurine, O-Phosphocholine, adenosine 5′-triphosphate (ATP), and glutamate (Appendix A). Additionally, samples from NC and NCDH, number 7 and 18 respectively, were separated from other replicates in the scores plot in the PC2 direction. Those samples, however, are within the 95% confidence interval of each group. Thus, we decided to include the samples clustered in other hierarchical groups as not all the CDH have the same severity. The loadings plot and a hierarchical clustering heat map (Appendix A) suggested that the underlying cause of the two samples being away from other samples is the high abundance of propylene glycol in sample 7 (NC) and low abundance of taurine in sample 18 (NCDH). Therefore, propylene glycol and taurine were excluded from the rest of the analysis.

Our next step was to perform partial least square discriminant analysis (PLS-DA) since it is an efficient and optimal method used in metabolomics when the number of metabolites detected is high and likely correlated. PLS-DA showed a well-defined separation between groups, being component 1 (50.6%) and component 2 (10.5%) (Figure 1A). Metabolites with variable importance in projection (VIP) scores greater than 1.0 significantly contribute to the model (Figure 1B). The metabolites with the highest VIP-score (>2) were lactate, glutamate, and ATP. This classification model had an accuracy of 0.83, Q2 of 0.59, and R2 of 0.84 for two components (Figure 1C). The top three classification variables identified in PLS-DA were also determined to be significant: lactate (*p* < 0.001), glutamate (*p* < 0.001) in VC vs. NC and NCDH, ATP, and O-phosphocholine (*p* < 0.001) in all pairs (Figure 1D).

Furthermore, univariate analysis was performed on all 64 metabolites in a pair-wise Student’s *t*-test (Appendix A). Out of 62 metabolites (taurine and propylene glycol were excluded), 44 metabolites were significantly altered (false discovery rate (FDR) *p* < 0.05) in one or more pair-wise comparisons (Table 1).

### 2.2. Effect of CDH in Fetal Lung Metabolism (NCDH vs. VC)

In order to investigate the molecular dysfunction under CDH, we compared the metabolomics profile in the newborn rats with normal lungs from VC with NCDH lungs. The 40 significantly altered metabolites were identified by Student’s *t*-test (Table 1). We conducted a volcano plot analysis (combine a fold change analysis (FC) and a *t*-test with FDR correction). We identified nine metabolites that were elevated and ten that were decreased (FC > 2.0 and *p* < 0.05 with FDR) (Figure 2A, Table 1).

The PLS-DA scores plot showed complete separation in component 1 (66.9%) (Figure 2B) along with the metabolites with the highest VIP-score (>2) of lactate, glutamate, ATP, and O-phosphocholine (Figure 2C). The cross-validation showed an accuracy of 1, Q2 of 0.93, and R2 of 0.99 for two components.

Multivariate exploratory Receiver Operating Characteristic (ROC) curve analysis was performed to identify the metabolites as potential biomarkers to distinguish NCDH lung from VC. The evaluation of different biomarker models’ performance revealed that the three selected metabolite models showed the predictive accuracy of 95% area under the curve (AUC) 0.98, with a confidence interval of 0.613–1 (Figure 2D). The results suggested that metabolites with the biomarker potential of NCDH are O-acetylcarnitine, AMP, ADP, UDP, UMP, and niacinamide (Figure 2E,F). Since the O-acetylcarnitine’s elevation was observed in NCDH and NC, it is not suitable to be a unique marker for the NCDH group. On the other hand, the changes of AMP, ADP, UDP, UMP, and niacinamide were unique to NCDH and different from VC and NC. These metabolites have the potential to be a biomarker for the NCDH group.

### 2.3. Effect of Lung Compression on the Fetal Lung Metabolism (NCDH vs. NC)

To differentiate the effect of the CDH from the exposure to nitrofen, we analyzed the differences between NCDH and NC groups, as both experimental groups of fetal rats were exposed to nitrofen. In this way, we exclude the effect that the teratogen will produce in the fetuses and focus on the impact of the mechanical effect of visceral herniation compressing the fetal lungs in development.

The comparisons between the two groups’ metabolomic profiles in the fetal lung tissue (NCDH-NC) were conducted in the same manner as NCDH-VC. The Student’s *t*-test identified 21 metabolites with statistically significant alterations (Table 1). The volcano plot analysis (FC > 2.0 and *p* < 0.05 with FDR) further identified five metabolites that were decreased (ADP, ATP, NAD, UDP, and 2-Aminobutyrate) and one that was increased (inosine).

The PLS-DA model showed a complete separation between the two groups in the component 1 direction with 31.5% variance (Appendix A). The metabolites with the highest VIP-score (>2) and loading factors were lactate, glutamate, ATP, and O-phosphocholine (Appendix A), which are similar to the NCDH-VC comparison. Cross-validation showed an accuracy of 0.92, Q2 of 0.67, and R2 of 0.94 for two components.

We compared NC and VC metabolic profiles to explore the nitrofen teratogenic effect on rat lung metabolism. We identified 27 metabolites using the Student’s *t*-test. The volcano plot analysis (FC > 2.0 and *p* < 0.05 with FDR) identified five elevated metabolites (uridine, O-acetylcarnitine, Inosine, Uracil, and UDP-galactose) and two decreased metabolites (ATP and Creatinine phosphate) (Table 1).

The two groups were separated clearly in the PLS-DA scores plot with 54.6% variance in Component 1 (Appendix A). The same set of metabolites were with a high VIP-score (>2) with previous comparisons: lactate, glutamate, ATP, O-phosphocholine (Appendix A). Cross-validation showed an accuracy of 1, Q2 of 0.82, and R2 of 0.95 for two components.

### 2.4. Metabolic Pathways Analysis

Metabolic pathways affected in each group were explored based on the sets of altered metabolites identified. Forty-eight pathways were identified as affected in NCDH compared with VC, and nineteen pathways met our criteria of impact > 0.1 and FDR *p* < 0.05 (Figure 3). The pathways affected are related to changes in oxidative stress (changes in the metabolism of nicotinate, nicotinamide, ascorbate, aldarate, and glutathione), nucleotide synthesis (pyrimidine metabolism, aminoacyl-t-RNA biosynthesis, amino-sugar, and nucleotid sugar metabolism), amino acid metabolism (glycine, serine, threonine, alanine, aspartate, Glutamate, glutamine, histidine, arginine, proline metabolism), glycerophospholipid metabolism, and glucose metabolism (pyruvate, (Tricarboxylic Acid) TCA cycle metabolism, and starch and sucrose metabolism).

In comparison to NC, NCDH showed nine pathways that met the criteria (Appendix A). These pathways were similar to NCDH vs. VC, including nicotinate and nicotinamide metabolism, pyrimidine metabolism, and the TCA cycle. Nineteen metabolic pathways were shown to be affected in NC compared with VC (Appendix A). The major effects were seen in various amino acid metabolism (glycine, serine, threonine, alanine, aspartate, glutamate, glutamine, histidine, arginine, proline, cysteine, and methionine). Oxidative stress regulation, nucleotide metabolism, and glycerophospholipid metabolism are also included in the list.

## 3. Discussion

The CDH lungs are hypoplastic with decreased DNA and protein content [29,30]. They have structural alterations on the pulmonary vessels characterized by a reduction in the arterial branching and increase media layer thickening in the small acinar and intra-acinar arterioles, causing PH [31,32,33]. Despite the exhaustive research, the pathophysiological changes during CDH development in-utero are still not well understood. To our knowledge, this study provides the first non-target metabolic analysis by NMR of fetal lung tissue with CDH in rat pups. Our study brings insights into the metabolic signatures of lung compression in CDH, lung hypoplasia, and PH in nitrofen-treated animals, with the potential of developing new therapeutic targets. The fetal lung metabolic tissue analysis is particularly attractive to study because it is selective of the occurrence in the lung cells without confounding effects from other organs’ signals.

Abnormalities in cellular metabolism cause multiple diseases, but metabolic dysfunction as a driver of respiratory pathology has been investigated only recently [28]. The four respiratory diseases with more research behind them are chronic obstructive pulmonary disease (COPD), asthma, pulmonary hypertension (PH), and idiopathic pulmonary fibrosis (IPF). Lung cellular metabolism is often discussed in the context of individual pathways. Still, it is essential to look at an integrated combination of metabolic pathways that allows the lung to function and thrive. Therefore, metabolic changes in one pathway cannot be fully understood in isolation without considering others.

Even though normal pulmonary development occurs in the uterus in a relatively hypoxic environment, compression in the fetal CDH lungs by the abdominal content could decrease the lung perfusion, causing a drop in the nutrients and oxygen delivery. Previous studies demonstrated that excessive prenatal lung hypoxia has significant and lasting effects on the pulmonary vasculature, structurally and functionally [34,35,36]. On the other hand, cells undergo anaerobic metabolism under hypoxia [37], putting them at risk of energetic failure if unable to produce enough ATP for cellular functions [38,39]. Therefore, cells increase the rate of anaerobic glycolysis by upregulating glycolytic enzymatic activity and decreasing oxidative phosphorylation. 

To study the CDH fetal lung metabolism, we used a well-established CDH animal model induced by the enteral administration of nitrofen (dichloro-1-(4-nitrophenoxy) benzene) in pregnant rats at E9.5 of gestation (Appendix A). The nitrofen model resembles human pathology, causing significant pulmonary hypoplasia, large diaphragmatic defects, cardiovascular abnormalities, and other malformations, with similar incidence as in human fetuses [40]. Moreover, the CDH-nitrofen model allowed us to analyze the lungs from the pups exposed to nitrofen without the diaphragmatic defect and observe the effects on the critical period of organogenesis [41]. Keijer et al. described the “dual hit” hypothesis explaining bilateral lung hypoplasia in the nitrofen model of CDH. It is hypothesized that a first insult affects both lungs before diaphragm development, followed by a second insult affecting the ipsilateral diaphragmatic defect lung driven by CDH-induced compression [42]. 

The results strongly suggested the significant alteration in glycolytic energy metabolites, antioxidant metabolites, and nucleotide metabolites in the NCDH group. Although some of the metabolic changes were commonly observed in the NC group, the alteration’s degree differed between the two groups.

The detection of significant accumulations of glucose and lactate and decreases in TCA cycle metabolites, fumarate, and citrate, in the CDH-nitrofen model strongly suggested the alteration in glycolytic energy metabolites (Figure 4).

Although all nitrofen-treated animals showed an increase in lactate compared to control, further induction of the lactate elevation in nitrofen-CDH was observed in this study. The lactate production in the lung is higher than in other tissues [34]. One potential explanation is that the lung has evolved to utilize aerobic glycolysis to decrease the local oxygen consumption to deliver more oxygen to other tissues. Additionally, lactate production can be used as an energy source for cells with short access to nutrients in the pulmonary circulation. Multiple studies have found an elevation of lactate in PH [27,28,29], indicating an increased glycolysis rate. ATP is the cell’s energy currency, and it is necessary to activate the ATP-dependent potassium channel (KATP) for modulation of the vascular tone [43,44]. Decreased potassium channel activity is a potential pathological substrate for PH in CDH [43]. The increase in lactate and the significant depletion of ATP in NCDH suggested suppressing glucose oxidative phosphorylation and the increase in glycolysis. Interestingly, the same metabolic profile was found by Marwan et al. after doing a fetal tracheal occlusion in rabbit fetuses without CDH [45,46].

A decrease in glucose utilization through glycolysis is suggested by the increase in glucose in the NCDH group. The number of glucogenic amino acids that feed into pyruvate and the TCA cycle metabolites (α-ketoglutarate, succinyl Coenzyme A, fumarate, or oxaloacetate) was decreased. Furthermore, reduced abundance of ketogenic amino acids, leucine, and lysine, and the branched-chain amino acids, valine, leucine, and isoleucine, were also observed in the NCDH group. Together, our data suggested that an increased amount of amino acids is being utilized to feed into glycolysis and the TCA cycle, possibly due to decreased glucose utilization. The increase in consumption of amino acids may also fuel the TCA cycle to increase ATP production. Amino acids are fundamental for synthesizing proteins, lipids, and nucleic acids [47], and are necessary for lung development [48]. The changes in amino acid concentrations in CDH have been reported previously in the tracheal aspirate of five newborns with CDH [49] and the metabolomic analysis in the amniotic fluid of 22 fetuses with CDH [50]. The compressive effect of the CDH affects the metabolism of some amino acids and nucleotides. It produces glucose metabolism changes (starch and sucrose, galactose, TCA cycle) and membrane and surfactant synthesis (glycerophospholipid metabolism). The lung compression following the early hypoplasia (as described in the “dual hit” hypothesis) caused a worse metabolic profile [42].

The fetal CDH metabolomic profile has changes associated with the lung hypoplasia caused by nitrofen and aggravated by lung compression. The CDH lungs have redox imbalance, bioenergetic failure, and an alteration in nucleotide and amino acid metabolism. The decrease in NAD^+^, NADP^+^, glutathione, and ascorbate leads to an increase in reactive oxygen species (ROS), especially H_2_O_2_, causing DNA damage [51,52]. Depending on the extent of the DNA damage, the cell can go from senescence to apoptosis or necrosis [53,54], explaining the lung hypoplasia found in PH and CDH. We found decreased metabolites involved in redox control (nicotinamide adenine nucleotide phosphate (NADP^+)^, reduced glutathione (GSH), and ascorbate). It is important to notice the severe depletion of NAD^+^ and NADP^+^. The oxidation of NADPH and GSH is necessary for ascorbate to reduce hydrogen peroxide (H_2_O_2_) to water, decreasing the oxidative stress (Figure 5).

The endogenous redox state affects the signaling pathways that regulate cell proliferation and differentiation, and it is critical during the fetal period when there is rapid tissue growth [55]. Glutathione is one of the essential antioxidants of embryonic development and exists in a couple of forms: oxidized (GSSG) and reduced (GSH) [55]. Glutathione synthesis requires ATP and three amino acids: glutamate, glycine, and cysteine [56]. Serine is also involved in the glutathione synthesis via the folate cycle, contributing to the glutathione reduction by intersecting with the methionine cycle [57].

Ascorbate (vitamin C) is an antioxidant molecule used as the electron donor to reduce H2O2 to water [58] and is necessary for collagen synthesis. High doses of ascorbate can increase glutathione in animals [59]. Glutathione and ascorbate spare each other, and their concentration remains stable if enough NADPH exists [60]. In the NCDH group, administration of a combination of antioxidants (vitamin A, C, E) on days E16–18 improved heart hypoplasia [61]. Ascorbate is decreased in NCDH compared with NC and VC (Table 1 and Figure 5).

Niacinamide or nicotinamide (vitamin B3) is the precursor for NAD^+^ and NADP^+^ and can be synthesized from the essential amino acid tryptophan. Nicotinamide inhibits lipid peroxidation and protein oxidation [62]. This vitamin can decrease ischemia/reperfusion-induced oxidative stress lung injury by providing a pool of NAD^+^ through the salvage pathway [63,64]. Nicotinamide is increased in NCDH compared with VC and NC, probably in an attempt to produce more NAD^+^ (Figure 5).

An increase in ROS production in the NCDH group has been described before [65,66]. Our metabolomics data add new evidence about the involvement of the redox homeostasis in the CDH pathophysiology.

Nucleotides are essential for tissue growth and cell replication. Nucleotides are part of coenzymes, energy currency (Guanosine Triphosphate (GTP), ATP), involved in biosynthesis (UDP for carbohydrates and CDP for lipids), and they are the second most important messengers in signaling pathways (e.g., cyclic AMP and cyclic Guanosine monophosphate (c-GMP)). In the CDH-nitrofen model, the decrease in the nucleotide triphosphates, ATP and GTP, and nucleotide diphosphates, UDP and ADP, were detected. In contrast, in the nucleotide monophosphate, AMP and UMP were increased along with their degradation products, uridine and inosine (Figure 6). AMP accumulation activates the adenine nucleotide degradation into IMP and inosine and posteriorly to hypoxanthine, xanthine, and uric acid. UDP is produced by phosphorylation using ATP as a phosphate donor. A decrease in ATP leads to the degradation of UMP to uridine. It appears that there is an increase in the nucleotide catabolism products with a reduction in the nucleotides that can provide energy (GTP, ATP).

There are two pathways for nucleotide synthesis, de novo and salvage. De novo synthesis is energetically expensive and critical for embryonic and fetal tissue [67]. The salvage pathways recover nucleosides and bases from exogenous sources and RNA and DNA degradation [68] to convert them into nucleotides for cells at rest and is an important energy-saving mechanism. The catabolic pathways for the excretion of nucleotides are fundamental to limit the toxic effect of nucleotides’ accumulation in the cells.

Based on the pair-wise comparisons (NCDH vs. VC), several key metabolites with the potential to be biomarkers to distinguish the NCDH group from the VC group were identified: AMP, ADP, UDP, UMP, and niacinamide. ATP and ADP are severely reduced, and AMP is increased in NCDH compared with NC and VC (*p* < 0.05). The abundance of three metabolites regulated in feedback mechanisms by AMP-Activated Protein Kinase (AMPK) is effective during the hypoxic condition [69,70]. Like AMP and ADP, the NCDH group showed an increase in UMP and decreased UDP. The change in the ratio of nucleotide monophosphate to diphosphate is a finding that will require more studies in the future.

Nitrofen, as described previously, causes lung hypoplasia, some degree of PH [71], and half of the litter develops CDH [72,73]. In this study, the effect of nitrofen in the lung metabolic profile (NC vs. VC) was mainly found in the amino acid metabolic pathways. For example, the decrease in lysine was observed similarly in both NC and NCDH. In contrast, the reduction in alanine and glutamate was further decreased, influenced by the CDH development. On the other hand, increased serine and threonine were reversed by the CDH development compared to VC. Nitrofen also causes placenta hypoplasia [74] and decreases the main retinol-binding protein (RBP) that transports vitamin A to the fetus. The changes in the metabolism of the amino acids may be due to the reduced transport of amino acids from the placenta to the fetus. However, no studies have successfully demonstrated this, to our knowledge. Other potential causes might be consuming in the TCA cycle metabolites, biosynthesis of metabolites, or used to produce glutathione. The nucleotide metabolism pathways (pyrimidine metabolism, t-RNA biosynthesis, nucleotide sugar metabolism) are affected. We expect to have an alteration in the nucleotide synthesis and degradation because the lungs are hypoplastic. Furthermore, the glycerophospholipid and amino sugar metabolisms are dysregulated, both important in the cellular membrane synthesis and expected in the lung with hypoplasia. Ascorbate, aldarate, and glutathione metabolism are affected, indicating dysregulation of the redox state in the fetal lungs with CDH.

More studies on CDH lung metabolism using other animal models and humans are needed to confirm our results. Based on our findings, it appears that CDH lungs have mitochondrial dysfunction. The mitochondria rely on the presence of ROS for internal regulation and signaling [75,76]. An increase in ROS causes mitochondrial derangements, leading to mitophagy and impairing ATP production [77]. Additional studies on the redox status and cellular bioenergetics focusing on the electron transport chain and the mitochondrial membrane potential can help to understand the changes described in our model. The metabolite abundance gives us information about the pathways affected, but it is impossible to know the changes in the efflux and influx responsible for the metabolite changes. For future studies, stable isotop resolved metabolomics (SIRM) using ^13^C-labeled glucose will be a useful way of tracing the flow through the pathways, allowing us to know the biosynthetic source of a metabolite and its catabolism.

Following our results, antioxidants (Vitamin C and NAD^+^ supplements) and other ROS scavengers could be useful to improve the cellular redox status and possibly the cellular bioenergetics and normal proliferation in the lung of CDH.

## 4. Materials and Methods

This study was performed using 18 timed-pregnant Spragu Dawley rats, 4–5 months old and weighing 200–250 g (Charles River Laboratories, Inc, Wilmington, MA. USA). They were housed individually at 22 °C on a standard dark:light schedule (10:14 h, i.e., light 7:00–19:00) with ad libitum access to water and standard chow.

Pregnant rats were divided into two groups: (1) vehicle control (VC) and (2) nitrofen. The mating date was considered E1, and the vaginal plug day was 0. On E9.5, pregnant rats received 100 mg of nitrofen dissolved in 1 mL of olive oil intra-gastrically, or received only olive oil as a vehicle control group. Pregnant rats were euthanized, and before pups were delivered by c-section on day E21.5 for samples harvest.

### 4.1. Tissue Collection

The fetal diaphragm was inspected for the presence of CDH at harvest. Total lung was harvested to maximize the number of metabolites extracted. Tissue sample acquisition for metabolomics was considered to avoid confounding biological interpretations of the data [78,79,80,81,82]. The pups in the nitrofen group were classified as nitrofen-CDH (NCDH), if they have CDH, or nitrofen control (NC) if CDH was not present. The three groups analyzed were nitrofen CDH (NCDH), nitrofen control (NC), and vehicle control (VC) (Appendix A).

A trained team of 4 people performed the total lung harvest of all the pups in 10–15 min to minimize hypoxia. The whole lung was carefully excised, minimizing surgical trauma following sample acquisition recommendations [78,79,80,81,82]. Tissue lung samples for metabolomics were harvested and immediately snapped frozen and stored at −80 °C until use. 

### 4.2. NMR Sample Preparation

The water content of lung tissue was determined before this experiment, which was 81.7%. The modified Bligh and Dyer extraction [80,83] was used to obtain polar metabolites. Briefly, cold methanol and water were added to the samples in tubes with 2.8 mm ceramic beads and homogenized for 30 s at 5000 rpm. Cold chloroform and water were added to the lung tissue to bring the final methanol:chloroform:water ratio to be 2:2:1.8. The polar phase was dried under vacuum and resuspended in 220 µL of NMR buffer containing 100 mM phosphate buffer (pH 7.3), 1 mM TMSP (3-Trimethylsilyl 2,2,3,3-d4 propionate), and 1 mg/mL sodium azide prepared in deuterium oxide (D_2_O). The final volume of 200 µL of each sample was transferred into a 3 mm NMR tube Bruker Avance III HD (Bruker Analytik, Rheinstetten, Germany) for data collection.

### 4.3. Spectra Acquisition

One dimensional (1D) ^1^H NMR spectra were acquired on a Bruker Avance III HD 600 MHz spectrometer with a 5 mm Broad Band Observed (BBO) Prodigy probe. All data were collected at a calibrated temperature of 298 K using the noesygppr1d pulse sequence in the Bruker pulse sequence library. All the data collection and processing were performed using Topspin 3.6 software (Bruker Analytik, Rheinstetten, Germany). For a representative sample, two-dimensional (2D) data ^1^H-^1^H total correlation spectroscopy (TOCSY) and ^1^H-^13^C heteronuclear single quantum coherence (HSQC) were collected for metabolite assignments. 

### 4.4. Metabolites’ Assignments and Quantification

Chemical shifts were assigned to metabolites based on 1D ^1^H, 2D TOCSY, and HSQC NMR experiments with reference spectra found in databases, Human Metabolome Database (HMDB) [84], and Chenomx^®^ NMR Suite profiling software (Chenomx Inc. version 8.4). The metabolites’ concentrations in polar extracts were calculated using Chenomx software based on the internal standard, TMSP. A total of 64 polar metabolites were assigned and quantified. Metabolite concentrations were normalized to the total spectral intensity for statistical analysis.

### 4.5. Statistical Analysis

Metabolite abundance (µmoles) was normalized to the total spectral intensity (TSI) before all statistical analyses. The required sample size and processing protocols did not permit accurate tissue weight measurements without loss and sample degradation. RNA quantifications had larger variability within the group and potential outliers, however, they had a similar trend with the data normalized to TSI. Normalized metabolite abundances were compared by pair-wise Student’s *t*-test using R studio.

All the multivariate analyses were performed in MetaboAnalyst 4.0. The Pareto scaling was applied to the normalized metabolite data to perform untargeted principal component analysis (PCA) and targeted partial least squares discriminant analysis (PLS-DA). The biomarker analysis was performed to obtain the ROC curves using the Linear Support Vector Machines (SVM) classification method and SVM built-in feature ranking method. Cross-validation was further performed on all models generated. 

### 4.6. Pathway Analysis

In order to identify the most relevant pathways affected by CHD, the pathway analysis in MetaboAnalyst 4.0 was performed using the following parameters: the global test method for the enrichment analysis, the relativ betweenness centrality for the topology analysis, and the *Rattus norvegicus* pathway library for the Kyoto Encyclopedia of Genes and Genomes (KEGG) database. A cutoff value of 0.1 for pathway impact and false discovery rate (FDR) approach *p* < 0.05 was selected to filter the less important pathways

## 5. Conclusions

There is a close relationship between oxidative stress, energy metabolism, cell survival, and proliferation. These changes can only be analyzed in a whole tissue context and not at the individual cellular level. Our metabolomic investigation of complete fetal lungs indicated the presence of a unique metabolic profile in nitrofen-induced CDH fetal lungs, different from nitrofen-controls and vehicl controls. These differences indicate evident changes in energy production, redox control state, and cell proliferation that impact normal lung development.

Future studies focusing on validating these results in other CDH animal models and new studies on energy failure and redox potential will give us new insights to target new therapies to improve the CDH patients’ clinical management.

## Figures and Tables

**Figure 1 metabolites-11-00177-f001:**
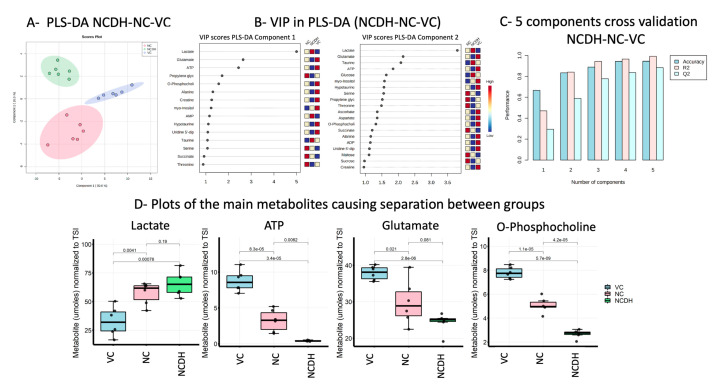
CDH fetal lungs’ metabolic profile. (**A**) Loading score plot of partial least square discriminant analysis (PLS-DA) comparing vehicle control (VC: blue), nitrofen control (NC: red), and nitrofen-CDH model (NCDH: green) with component 1 (Explained variance (EV) 50.6%), component 2 (EV 10.5%), and 95% confidence interval, and (**B**) the corresponding PLS-DA metabolites variable importance in projection (VIP) from component 1 and 2. (**C**) 5 components cross-validation assessment of the PLD-DA model showing the accuracy of 0.83, Q2 of 0.59, and R2 of 0.84 for 2 components. (**D**) Box plots of the selected altered metabolites between the treatment groups with *p*-values based on the Student’s *t*-test: VC—blue, NC—red, and NCDH—green.

**Figure 2 metabolites-11-00177-f002:**
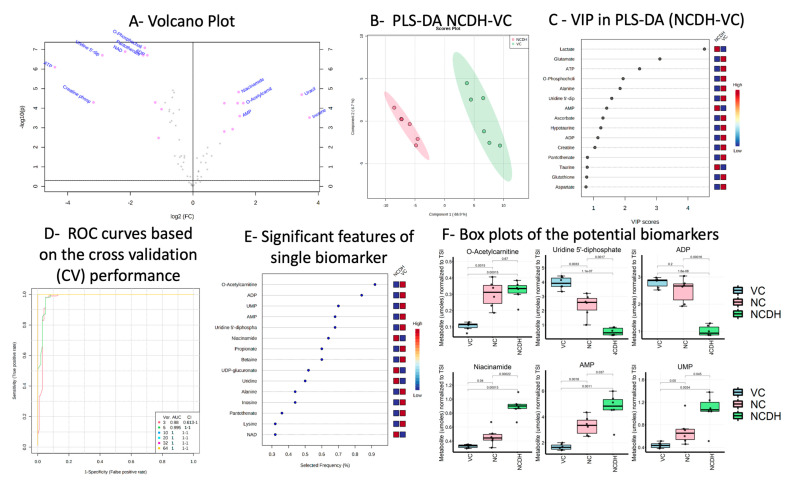
Comparison between NCDH-VC and biomarker analysis. (**A**) Volcano plot analysis comparing 62 metabolites between nitrofen-CDH (NCDH) and vehicle control (VC), highlighting nine increased and decreased metabolites (FC > 2.0 and *p* < 0.05). (**B**) Loading score plot of partial least square discriminant analysis (PLS-DA) of VC (green) and NCDH (red) with component 1 (Explained Variance (EV) 66.9%), component 2 (EV 8.7%), and 95% confidence interval, and (**C**) the corresponding PLS-DA metabolites’ variable importance in projection (VIP) from component 1. (**D**) Receiver operating characteristic (ROC) curves for the predictive models of various sets of biomarkers based on the average cross-validation performance. (**E**) The corresponding significant features of a single biomarker model ranked by importance. (**F**) Box plots of the potential biomarkers for CDH-nitrofen model with *p*-values based on the Student’s *t*-test: VC—blue, NC—red, and NCDH—green.

**Figure 3 metabolites-11-00177-f003:**
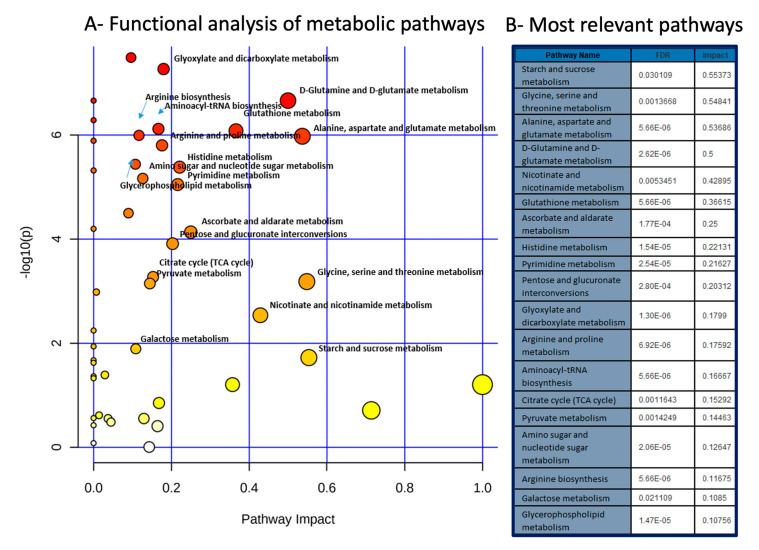
Functional and metabolic pathways analysis in nitrofen-CDH (NCDH) model. (**A**) Metabolome view map of relevant metabolic pathways for changes detected in NCDH compared to vehicle control (VC) based on the pathway enrichment and topology analysis. (**B**) Table of most relevant metabolic pathways with the FDR-corrected *p*-value (<0.05) and its impact factor (>0.1) from topology analysis.

**Figure 4 metabolites-11-00177-f004:**
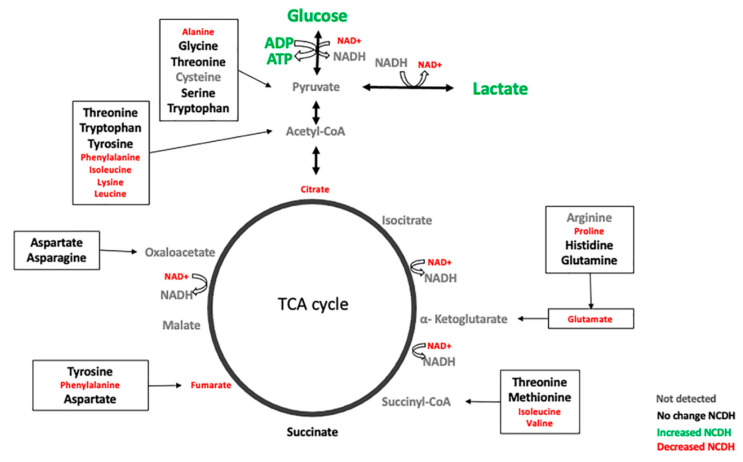
Modified glycolysis, TCA cycle, and amino acids “anaplerosis” in fetal CDH lungs. Schematic diagram of glycolytic, TCA cycle, and amino acids metabolites flow and the changes observed in the NCDH. Green (increased), red (decreased), black (no change), and grey (undetected).

**Figure 5 metabolites-11-00177-f005:**
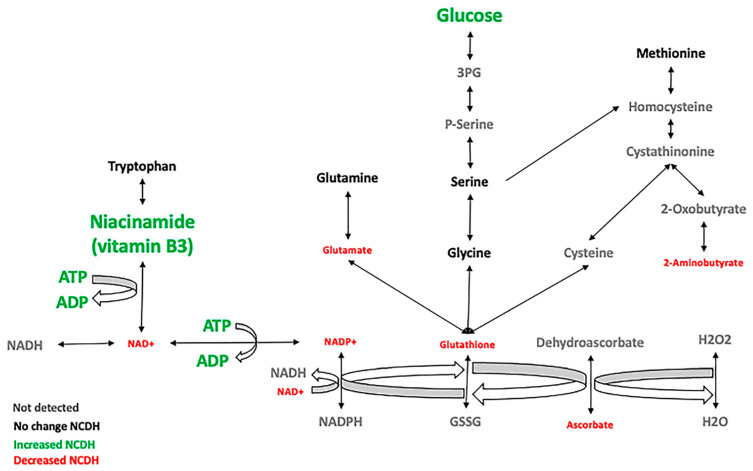
Decreased antioxidant levels in fetal CDH lungs. Schematic diagram of glutathione and niacinamide metabolism and changes identified in the NCDH group. Green (increased), red (decreased), black (no change), and grey (undetected).

**Figure 6 metabolites-11-00177-f006:**
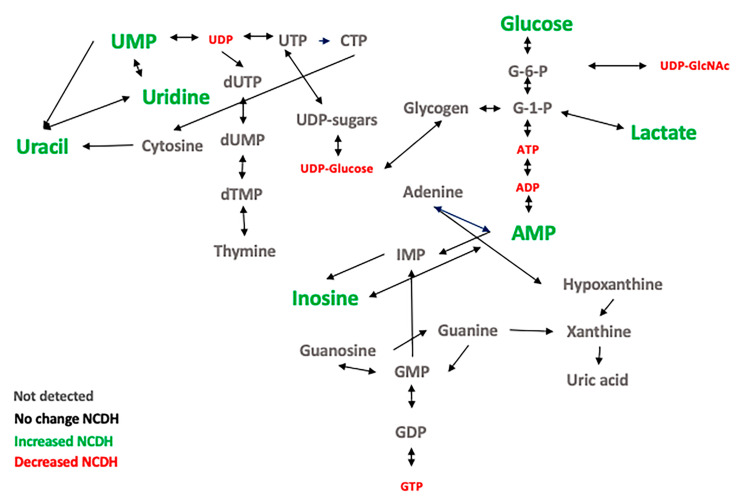
Alterations of nucleotide metabolism in fetal CDH lung. The overview of the metabolic flow of nucleotide metabolites and changes identified in the NCDH group compared with the VC group. Green (increased), red (decreased), black (no change), and grey (undetected).

**Table 1 metabolites-11-00177-t001:** Comparisons of metabolites detected in lung tissue samples using Hydrogen-1 Nuclear Magnetic Resonance (1H NMR) metabolomics. The 62 metabolite changes in each pair-wise comparison are listed with *p*-values based on the Student’s *t*-test with false discovery rate (FDR) correction (*p* < 0.05 in red). The fold changes are calculated by NCDH/NC, NCDH/VC, and NC/VC, where the fold change > 1 indicates the increase and <1 indicates the decrease compared to the control group. If the fold change is <1, the negative reciprocal is listed.

	NCDH-VC		NCDH-NC		NC-VC			NCDH-VC		NCDH-NC		NC-VC	
Metabolites	*p*-Value	Fold Change	*p*-Value	Fold Change	*p*-Value	Fold Change	Metabolites	*p*-Value	Fold Change	*p*-Value	Fold Change	*p*-Value	Fold Change
2-Aminobutyrate	1.91 × 10^−4^	−2.30	3.28 × 10^−2^	−2.20	8.24 × 10^−1^	−1.05	Lysine	4.51 × 10^−5^	−1.58	9.46 × 10^−1^	−1.01	9.37 × 10^−3^	−1.56
3-Hydroxybutyrate	5.14 × 10^−2^	1.09	7.84 × 10^−2^	1.16	3.93 × 10^−1^	−1.06	Maltose	1.37 × 10^−1^	1.37	5.70 × 10^−1^	−1.26	2.64 × 10^−1^	1.74
Acetate	6.02 × 10^−2^	1.19	9.29 × 10^−1^	1.01	9.44 × 10^−2^	1.18	Mannose	3.25 × 10^−1^	1.20	7.84 × 10^−2^	−1.45	2.25 × 10^−2^	1.74
ADP	3.38 × 10^−7^	−2.74	3.37 × 10^−3^	−2.47	2.75 × 10^−1^	−1.11	Methionine	8.56 × 10^−1^	1.02	5.70 × 10^−1^	−1.05	3.79 × 10^−1^	1.07
Alanine	3.36 × 10^−4^	−1.54	2.20 × 10^−2^	−1.28	4.28 × 10^−2^	−1.20	myo-Inositol	3.34 × 10^−1^	−1.06	4.71 × 10^−2^	1.20	1.51 × 10^−3^	−1.27
AMP	2.83 × 10^−3^	2.81	9.18 × 10^−2^	1.42	9.37 × 10^−3^	1.99	NAD^+^	2.20 × 10^−7^	−4.48	8.97 × 10^−3^	−2.99	1.62 × 10^−2^	−1.50
Ascorbate	2.33 × 10^−4^	−1.69	4.23 × 10^−3^	−1.58	2.58 × 10^−1^	−1.07	NADP^+^	5.00 × 10^−3^	−1.47	2.06 × 10^−2^	−1.41	7.30 × 10^−1^	−1.04
Asparagine	6.34 × 10^−1^	−1.09	1.50 × 10^−1^	−1.31	1.71 × 10^−1^	1.20	Niacinamide	4.75 × 10^−4^	2.75	3.44 × 10^−3^	1.93	7.97 × 10^−2^	1.42
Aspartate	9.99 × 10^−2^	−1.30	1.78 × 10^−1^	−1.22	6.92 × 10^−1^	−1.07	O-Acetylcarnitine	4.99 × 10^−4^	3.05	7.38 × 10^−1^	1.06	9.37 × 10^−3^	2.87
ATP	1.98 × 10^−4^	−21.21	2.78 × 10^−2^	−7.84	1.06 × 10^−3^	−2.70	O-Phosphocholine	2.20 × 10^−7^	−2.90	1.62 × 10^−3^	−1.89	5.88 × 10^−4^	−1.54
Betaine	1.05 × 10^−2^	1.38	3.43 × 10^−2^	1.37	9.53 × 10^−1^	1.01	O-Phosphoethanolamine	3.17 × 10^−1^	−1.13	9.83 × 10^−1^	−1.00	2.54 × 10^−1^	−1.13
Butyrate	7.98 × 10^−2^	−1.39	3.94 × 10^−1^	−1.20	2.58 × 10^−1^	−1.16	Pantothenate	8.99 × 10^−6^	−3.00	1.62 × 10^−3^	−1.90	5.88 × 10^−4^	−1.58
Choline	8.33 × 10^−4^	2.00	1.50 × 10^−1^	1.24	2.75 × 10^−2^	1.61	Phenylalanine	3.11 × 10^−2^	−1.30	4.48 × 10^−2^	−1.27	8.24 × 10^−1^	−1.03
Citrate	5.74 × 10^−4^	−1.60	7.84 × 10^−2^	−1.30	6.94 × 10^−2^	−1.24	Proline	6.02 × 10^−2^	−1.23	1.85 × 10^−1^	−1.25	9.20 × 10^−1^	1.02
Creatine	2.01 × 10^−4^	−1.35	3.94 × 10^−1^	−1.06	3.68 × 10^−3^	−1.28	Propionate	3.68 × 10^−2^	1.51	2.88 × 10^−1^	1.26	3.93 × 10^−1^	1.20
Creatine phosphate	8.12 × 10^−4^	−9.02	1.49 × 10^−1^	−2.69	3.68 × 10^−3^	−3.36	Serine	3.17 × 10^−1^	1.11	2.19 × 10^−1^	−1.13	1.26 × 10^−2^	1.25
Formate	6.53 × 10^−5^	−1.52	9.18 × 10^−2^	−1.24	4.90 × 10^−2^	−1.22	sn-Glycero-3-phosphocholine	2.16 × 10^−2^	1.36	2.06 × 10^−2^	1.40	8.24 × 10^−1^	−1.03
Fumarate	3.33 × 10^−2^	−1.34	5.70 × 10^−1^	−1.08	7.97 × 10^−2^	−1.24	Succinate	6.62 × 10^−1^	1.07	2.78 × 10^−2^	−1.66	1.62 × 10^−2^	1.77
Glucose	6.41 × 10^−2^	1.33	2.78 × 10^−2^	1.66	2.58 × 10^−1^	−1.25	Sucrose	8.70 × 10^−1^	−1.03	1.76 × 10^−1^	−1.72	1.99 × 10^−1^	1.67
Glutamate	3.00 × 10^−5^	−1.56	1.50 × 10^−1^	−1.23	4.93 × 10^−2^	−1.27	Threonine	7.21 × 10^−1^	−1.04	2.20 × 10^−2^	−1.34	4.28 × 10^−2^	1.28
Glutamine	1.87 × 10^−1^	1.13	6.56 × 10^−1^	1.05	2.75 × 10^−1^	1.08	Tryptophan	9.84 × 10^−1^	−1.00	1.85 × 10^−1^	−1.21	2.31 × 10^−1^	1.20
Glutathione	3.11 × 10^−2^	−1.49	5.14 × 10^−1^	−1.14	7.19 × 10^−2^	−1.30	Tyrosine	8.56 × 10^−1^	−1.02	9.83 × 10^−1^	1.00	8.24 × 10^−1^	−1.02
Glycine	1.99 × 10^−1^	1.13	3.58 × 10^−1^	1.12	9.49 × 10^−1^	1.01	UDP-galactose	8.75 × 10^−3^	1.82	2.19 × 10^−1^	−1.23	5.74 × 10^−3^	2.24
GTP	3.68 × 10^−4^	−2.00	6.87 × 10^−3^	−1.81	2.59 × 10^−1^	−1.11	UDP-glucose	8.12 × 10^−4^	−1.66	1.50 × 10^−1^	−1.28	4.59 × 10^−2^	−1.30
Histidine	6.00 × 10^−1^	1.18	9.29 × 10^−1^	1.04	5.51 × 10^−1^	1.13	UDP-glucuronate	3.11 × 10^−2^	1.26	6.56 × 10^−1^	−1.07	7.97 × 10^−2^	1.34
Hypotaurine	1.37 × 10^−1^	−1.26	5.70 × 10^−1^	−1.10	3.07 × 10^−1^	−1.15	UDP-N-Acetylglucosamine	2.01 × 10^−4^	−1.68	1.89 × 10^−1^	−1.14	1.06 × 10^−3^	−1.47
Inosine	3.99 × 10^−3^	13.18	2.77 × 10^−2^	2.88	4.88 × 10^−2^	4.58	UMP	7.34 × 10^−3^	2.40	1.04 × 10^−1^	1.52	9.44 × 10^−2^	1.58
Isobutyrate	5.69 × 10^−4^	−1.85	3.46 × 10^−2^	−1.43	4.80 × 10^−2^	−1.30	Uracil	2.33 × 10^−4^	11.11	9.33 × 10^−1^	1.03	2.25 × 10^−2^	10.76
Isoleucine	3.68 × 10^−2^	−1.21	1.29 × 10^−1^	−1.18	8.24 × 10^−1^	−1.02	Uridine	1.54 × 10^−4^	2.68	1.78 × 10^−1^	−1.18	5.88 × 10^−4^	3.16
Lactate	1.94 × 10^−3^	2.00	2.74 × 10^−1^	1.14	1.62 × 10^−2^	1.75	UDP	1.77 × 10^−6^	−7.42	1.39 × 10^−2^	−4.46	1.49 × 10^−2^	−1.66
Leucine	3.06 × 10^−2^	−1.21	9.62 × 10^−2^	−1.16	6.22 × 10^−1^	−1.04	Valine	5.41 × 10^−3^	−1.35	1.50 × 10^−1^	−1.18	2.06 × 10^−1^	−1.15

## Data Availability

The data presented in this study are available on request from the corresponding author. The data are not publicly available due to author’s preference.

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
