# Peer review of "Lung Metabolomics Profiling of Congenital Diaphragmatic Hernia in Fetal Rats"

_metabolites, 2021, doi:10.3390/metabo11030177_

Round 1

Reviewer 1 Report

Overall very interesting manuscript with new insights into the molecular mechanisms behind CDH pathophysiology. There were minor grammatical and spelling errors throughout the manuscript. In the opening paragraph, statistics were given without citations, would recommend adding additional citations when facts are stated. In the third introduction paragraph the authors write “hypoplastic CDH lungs have a decrease in the pulmonary flow, mostly to the left lung.” I am assuming this is for infants with left CDH but what about those with right, central, or ventral CDH, is this also the case? Consider clarifying this statement if it is specific to only left CDH.

Author Response

Dear Reviewer,

Thank you for your comments and for the opportunity given to us for clarifying some aspects of the prior submission. Please find attached a new version with the changes to the prior version in blue. You will also found a point-by-point answer to the prior criticism. 

Comment 1: There were minor grammatical and spelling errors throughout the manuscript

Authors’ Response:

Thanks so much for detecting grammatical spelling errors now corrected in the manuscript

Change to Text:

Changes and misspelling words are in blue for easy identification by the reviewers

Comment 2: In the opening paragraph, statistics were given without citations, would recommend adding additional citations when facts are stated.

Authors’ Response: Thanks so much for the suggestion. We totally agreed with the comments and a new citation is added in the manuscript supporting the statement.

Change to Text:

The CDH incidence is 1.93/10 000 births in North America, with an overall 45.89% mortality in the first year of life [2].

Comment 3: In the third introduction paragraph the authors write “hypoplastic CDH lungs have a decrease in the pulmonary flow, mostly to the left lung.” I am assuming this is for infants with left CDH but what about those with right, central, or ventral CDH, is this also the case? Consider clarifying this statement if it is specific to only left CDH

Authors’ Response: Thanks so much for the comment and the suggestion. Hypoplastic CDH lungs have a decrease in the pulmonary flow, being worse on the ipsilateral side and a new reference is added.

Change to Text:

Hypoplastic CDH lungs have a decrease in the pulmonary flow, being worse on the ipsilateral side.[22–25].

Reviewer 2 Report

In this paper the Authors aim to assess the Lung metabolomics profiling of congenital diaphragmatic hernia. A comprehensive and extensive literature review of the NCBI database PubMed was also carried out. The article was well conducted and it is interesting in its fields. It is a well-structured paper, written in good English and the References are up dated. 

Minor issues:

In the “discussion” section I suggest to understand the impact of reflux in a particular subset of subjects. Therefore, the following paper should be considered:

“Diagnostic Criteria for Gastro-esophageal Reflux Following Sleeve Gastrectomy. Lim G, Johari Y, Ooi G, Playfair J, Laurie C, Hebbard G, Brown W, Burton P. Obes Surg. 2021 Jan 25. doi: 10.1007/s11695-020-05152-5”

“Aiolfi A, Micheletto G, Marin J, Rausa E, Bonitta G, Bona D. Laparoscopic Sleeve-Fundoplication for Morbidly Obese Patients with Gastroesophageal Reflux: Systematic Review and Meta-analysis. Obes Surg. 2021 Jan 3. doi: 10.1007/s11695-020-05189-6. Epub ahead of print. PMID: 33389630.”

I consider that the paper could be published in the Journal after these revisions.

Author Response

Dear Reviewer,

Thank you for your comments and for the opportunity given to us for clarifying some aspects of the prior submission. Please find attached a new version with the changes to the prior version in blue. You will also found a point-by-point answer to the prior criticism. 

Comment 1: In the “discussion” section I suggest to understand the impact of reflux in a particular subset of subjects. Therefore, the following paper should be considered:

“Diagnostic Criteria for Gastro-esophageal Reflux Following Sleeve Gastrectomy. Lim G, Johari Y, Ooi G, Playfair J, Laurie C, Hebbard G, Brown W, Burton P. Obes Surg. 2021 Jan 25. doi: 10.1007/s11695-020-05152-5”

“Aiolfi A, Micheletto G, Marin J, Rausa E, Bonitta G, Bona D. Laparoscopic Sleeve-Fundoplication for Morbidly Obese Patients with Gastroesophageal Reflux: Systematic Review and Meta-analysis. Obes Surg. 2021 Jan 3. doi: 10.1007/s11695-020-05189-6. Epub ahead of print. PMID: 33389630.”

Author’s Response:

Thanks for the suggestion, we think clarifying that point in the introduction and adding the references are a great addition to the manuscript

Change to Text:

The CDH incidence is 1.93/10 000 births in North America, with an overall 45.89% mortality in the first year of life [2]. Half of the cases, approximately, are associated with other congenital malformations [3–5]. The survivors face considerable long-term morbidities, such as respiratory problems, nutritional issues, neurodevelopmental delays, hernia recurrence, and orthopedic complications needing a multidisciplinary approach [2,6]. Pathological Gastro-esophageal reflux disease and the presence of hiatal hernia in patients who have undergone CDH repair are both known long-term complications [7,8].

Reviewer 3 Report

In the current study, Maria del Mar Romero-Lopez et al. studied the metabolic changes of fetal lung upon induction of diaphragmatic hernia.

The manuscript is well written and the authors have used the standard and well defined model of nitrofen induced congenital diaphragmatic hernia in rats.

The metabolic profiling is conducted with state of the art methods and analysis. All the data look robust and well presented. As stated by the authors the metabolic changes observed could lay the ground for future biomarkers identification or targets for new treatment and are thus of great interest for the field.

I would recommend to accept the current manuscript after improvement of the resolution of the figures, especially improving the size and sharpness of the text of the graphs which are for most unreadable  

I would also advise to remove the border of figures 4 and 5 to stay consistent with the other figures of the manuscript.

Author Response

Dear Reviewer,

Thank you for your comments and for the opportunity given to us for clarifying some aspects of the prior submission. Please find attached a new version with the changes to the prior version in blue. You will also found a point-by-point answer to the prior criticism. 

Author’s Response:

Thanks so much for your review and for the suggestions, we agree with the reviewer that the figures needed to be in higher/better resolution. All the issue have been addressed and better figures uploaded in the manuscript.